# High quality factor metasurfaces for two-dimensional wavefront manipulation

Claudio U. Hail [1], Morgan Foley[2], Ruzan Sokhoyan [1], Lior Michaeli[1] & Harry A. Atwater [1] ✉

The strong interaction of light with micro- and nanostructures plays a critical role in optical sensing, nonlinear optics, active optical devices, and quantum optics. However, for wavefront shaping, the required local control over light at a subwavelength scale limits this interaction, typically leading to low-quality-factor optical devices. Here, we demonstrate an avenue towards high-quality-factor wavefront shaping in two spatial dimensions based on all-dielectric higher-order Mie-resonant metasurfaces. We design and experimentally realize transmissive band stop filters, beam deflectors and high numerical aperture radial lenses with measured quality factors in the range of 202–1475 at near-infrared wavelengths. The excited optical mode and resulting wavefront control are both local, allowing versatile operation with finite apertures and oblique illumination. Our results represent an improvement in quality factor by nearly two orders of magnitude over previous localized mode designs, and provide a design approach for a new class of compact optical devices.

The recirculation of light in a confined optical mode is a ubiquitous method to amplify the interaction of light and matter. In this respect, the ability to confine the light to the resonating mode is quantified by the quality factor, $Q$, as the energy stored per round-trip optical loss in the resonator. With optical micro- and nanostructures, including Fabry-Pérot cavities[1,2], whispering gallery mode resonators[3–6], photonic crystals[7,8], guided mode structures[9], and bound states in the continuum (BIC)[10–13], quality factors of up to $10^8$ have been demonstrated. The high level of field enhancement and confinement attained in these structures has led to many advances in sensing[14,15] active optical devices[16,17], light sources[11,18], and amplification of photon–matter coupling[2,19]. However, in general, as the mode volume of an optical resonator decreases and the mode becomes more localized, more radiative decay channels become available, and the field enhancement and quality factor diminish[20]. As a result, there is typically a tradeoff between spatial mode localization and the attainable quality factor.

In optical metasurfaces, a subwavelength-spaced array of localized resonators is used to abruptly manipulate the phase, amplitude, polarization, and spectrum of light at an interface[21]. These structures hold promise to revolutionize many areas of optical imaging, communication, sensing, and display technology. Numerous optical components and phenomena have been realized using metasurfaces such as efficient flat lenses[22–24], on-chip holography[25,26] and dynamic beam steering[27,28]. Attaining strong light–matter interaction, and hence high quality factors, in metasurfaces is particularly desirable, as it enables the realization of ultra-fast spatial light modulators, non-linear parametric conversion, highly responsive optical sensing, and tailored light emission. However, the required subwavelength scale wavefront control imposes a limit on the resonator size, leading to significant radiative loss. As a result, most metasurfaces have been broadband and have relied on dielectric structures with limited light confinement and hence quality factor ($Q < 15$)[23,24,29]. Only recently, advances in wavefront manipulation with increased quality factors have been made with structures relying on extended guided mode resonance[30–32] and nonlocal modes based on bound states in the continuum[33,34]. Notably, achieving simultaneous local control over a wavefront with resonance phase and high quality factor remains an outstanding challenge[35,36]. Here, we demonstrate an avenue towards high-quality factor metasurfaces by leveraging higher-order Mie resonances to manipulate the wavefront of light locally in two dimensions based on the resonance phase.

[1]Thomas J. Watson Laboratory of Applied Physics, California Institute of Technology, Pasadena, CA 91125, USA. [2]Department of Physics, California Institute of Technology, Pasadena, CA 91125, USA. ✉e-mail: haa@caltech.edu

## Results

Figure 1a illustrates our high-quality factor optical metasurface for wavefront manipulation in two dimensions. The surface consists of subwavelength-spaced amorphous silicon nanoblocks of length $L$ and height $H$ arranged in a square array with periodicity $P$ on a transparent glass substrate. The structure dimensions are in the sub-diffractive regime, both in air and in the substrate, to avoid exciting any lattice modes. The geometric parameters are chosen to induce and spectrally overlap an electric dipole (ED) and electric octupole (EO) mode in the nanoblocks at near-infrared wavelengths. With the ED and EO modes, the surface operates as a higher-order Mie-resonant metasurface enabling local control over the transmitted wavefront. Figure 1b shows the transmission and reflection spectrum of the metasurface with a uniform nanoblock side length $L = 555$ nm, as calculated with finite difference time domain (FDTD) simulations (see Methods for details). A multipole expansion of the modes inside a single nanoblock embedded in the array shows that the ED and EO modes are in phase and of similar strength on resonance (see Supplementary Note 1)[37,38]. The destructive interference of the ED/EO mode and the illumination induces a sharp dip in transmission and near-unity reflection on resonance. A quality factor of $Q = 290$ is obtained from a Fano resonance fit[39]. On resonance, the electric field in the resonator shows a strong field enhancement of more than 28 times the incident field amplitude (see electric and magnetic field profiles Supplementary Note 1). Furthermore, the EO mode is clearly visible in the electric and magnetic field profiles. Notably, interference of multiple extended[11] or local modes[13,40] has been a common strategy to achieve a high-quality factor optical response. Analyzing the light scattered from the metasurface with varying nanoblock aspect ratios suggests that the ED/EO mode is different from extended supercavity modes[11] or supercavity modes reported in isolated nanoparticles[13,40] (see Supplementary Note 2).

Due to the near-field coupling of the ED and EO among neighboring elements, the resonance is dependent on the array size, i.e., the number of repetitions of the unit cell, which is the signature of a partially delocalized mode. This array size dependence is similar to that seen in low-order Mie-resonant metasurfaces[29] or asymmetry-induced quasi-BIC structures[41]. Our calculations suggest that for an array size beyond $10 \times 10$ unit cells, there is no further significant change in the modal properties (see Supplementary Fig. 1). For the ED/EO mode, we attribute this increase in quality factor with increasing array size to an enhancement of the ED through multipole coupling of the ED and EO in the array through neighboring particles, as similarly observed in a different geometry previously[42]. To further provide insight into the ED/EO mode we study its degree of localization. Exciting the mode in a single nanoblock of the array shows that it is localized within the nanoblock but extends primarily to its nearest neighbors along the direction of polarization (see Supplementary Note 3). From this calculation, we determine a mode volume of 3.6 times the volume of a unit cell, or $0.86 \lambda^3$, where $\lambda$ is the wavelength in free space. The mode localization can also be inferred from its dependence on the incidence angle of the illumination (see Fig. 1d, e). The ED/EO modes can be excited at oblique incidence for both transverse electric (TE) and transverse magnetic (TM) polarization. We observe a spectral shift of the resonance of less than 2 nm for a 10° change in the incident angle. For comparison, a fully delocalized lattice mode (Wood's anomaly) or guided mode resonance at the same resonance wavelength would result in a spectral shift on the order of 130 nm for the same 10° change[9]. The observed flat angular dispersion (angular dependent resonance wavelength shift) of our surface indicates that the studied high-Q mode is localized. Furthermore, the dispersion is of similar order as reported in low-order Mie-resonant metasurfaces based on the spectral overlap of an electric and magnetic dipole[43,44].

The local response of our higher-order Mie-resonant metasurface enables local wavefront manipulation by controlling the phase of the transmitted light. Varying the length $L$ of the nanoblocks, spectrally shifts the ED and EO modes and hence the resonance wavelength. The spectral shift accompanying the change in length $L$ allows employing the resonance phase to impose a phase shift on the transmitted light.

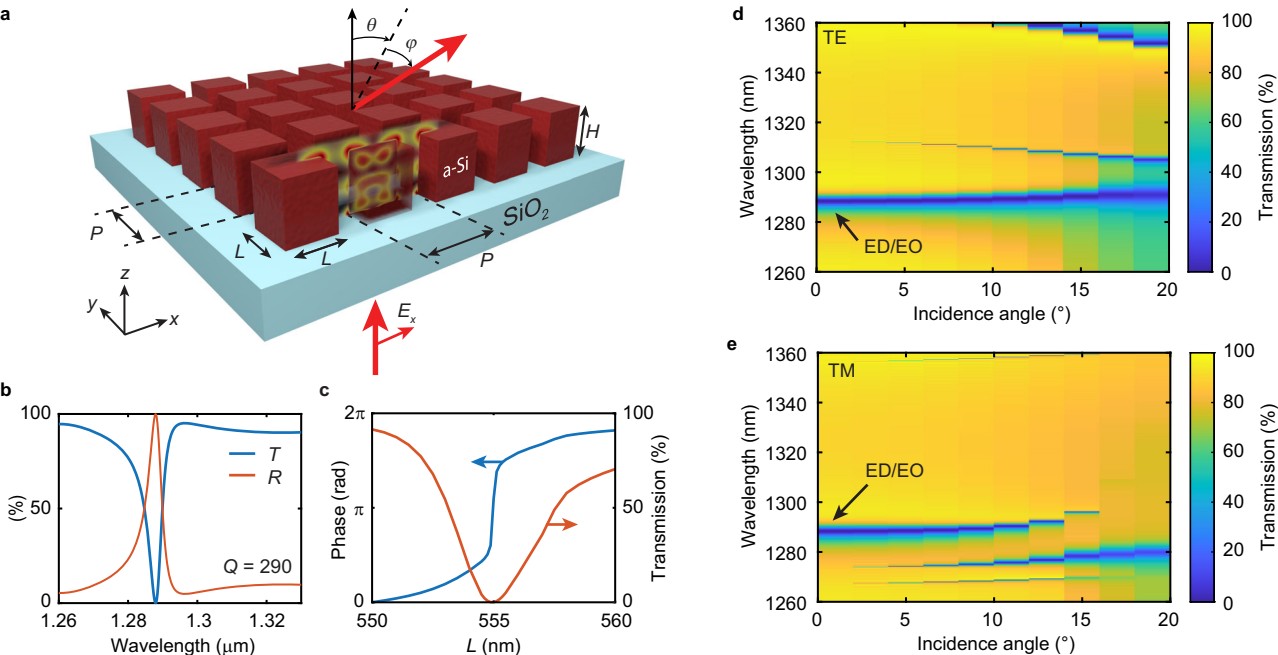

**Fig. 1 | High-quality factor metasurfaces for two-dimensional wavefront manipulation. a** Schematic of the metasurface consisting of amorphous silicon nanoblocks on a glass substrate. The surface is illuminated at normal incidence and can deflect light along the angles $\theta$ and $\varphi$ in the full upper hemisphere. **b** Calculated transmission ($T$) in blue and reflection ($R$) spectra in red of the metasurface with $P = 736$ nm, $H = 695$ nm, $L = 555$ nm. **c** Calculated transmission intensity (blue) and phase (red) with varying nanoblock side lengths $L$, $P = 736$ nm, and $H = 695$ nm at a wavelength of $\lambda = 1288$ nm. **d, e** Simulated transmission of the metasurface in (**a**) with varying angle of incidence for TE (**d**) and TM (**e**) polarization.

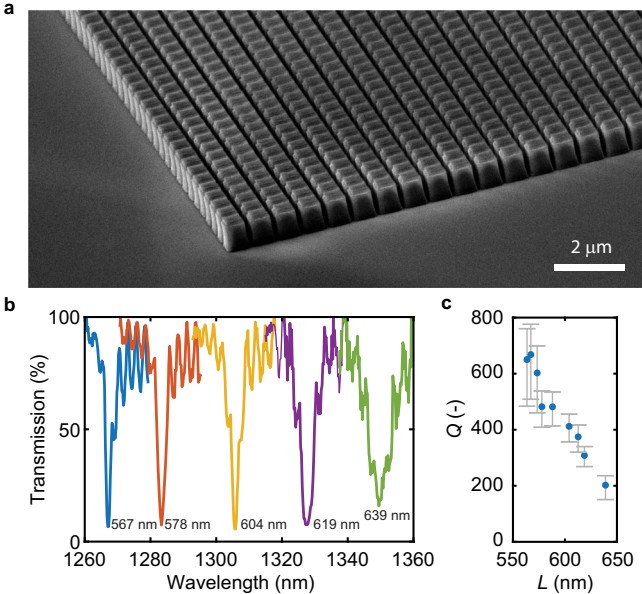

**Fig. 2 | High-quality factor resonances in silicon nanoblock arrays. a** Scanning electron micrograph of a metasurface with uniform nanoblock side lengths. **b** Experimentally measured transmission spectra of high-quality factor metasurfaces with varying nanoblock side lengths. The measured nanoblock side length in nm is indicated next to each curve. Here, $P = 736$ nm and $H = 695$ nm. **c** Experimentally measured quality factors with varying nanoblock side lengths as determined by a Fano lineshape fit. The error bars illustrate the 95% confidence interval of the fit. The metasurface size is 150 μm × 150 μm.

Figure 1c shows that varying the nanoblock side length by only ±1%, allows the phase of the transmitted light to be controlled over almost the entire 0–2$\pi$ range at a fixed wavelength of $\lambda = 1288$ nm. The inherent symmetry of the unit cell and the resonant mode also result in a polarization-independent response. Most remarkably, a small variation (≤1%) of the length $L$ by $dL$ of one block in an array of nanoblocks with uniform length $L$ manifests itself in a local resonance shift of the single nanoblock acting as an effective point source scatterer (see Supplementary Note 3). Furthermore, the resonant mode remains intact, and the high quality factor is preserved. These characteristics render our metasurface, which exhibits four-fold rotational symmetry and supports partially delocalized ED/EO modes, uniquely suitable for high-quality factor wavefront manipulation in two dimensions.

We fabricated our high-Q metasurfaces by single-step electron beam lithography and dry etching of plasma-deposited amorphous silicon on a glass substrate (see Methods for details). Figure 2a illustrates a scanning electron micrograph of a metasurface with uniform nanoblock side lengths. To experimentally characterize the surfaces, we employed linearly polarized, normally incident light through the substrate from a wavelength-tunable diode laser, collected the transmitted light with an objective lens and imaged it onto an InGaAs IR-camera, or focused it onto a power meter (see Methods for details). Figure 2b shows the measured transmission spectra of metasurfaces with uniform nanoblock side lengths. A strong dip in transmission is observed on resonance with a narrow linewidth ranging between 2 and 8 nm. On resonance, the measured minimum transmission ranges between 6–16%. The corresponding measured quality factors range between 202 and 668 as determined by fitting a Fano lineshape function. With uniform nanoblock sizes, this metasurface acts as an ultra-thin narrowband bandstop filter. An increase in nanoblock side length of only a few nanometers significantly redshifts the resonance by more than the linewidth. Furthermore, increasing the nanoblock side length beyond $L = 578$ nm leads to a decrease in the quality factor. The side length is thus a convenient parameter for adjusting the quality

factor by design. The high-frequency oscillation in the transmission spectrum is due to the interference of light reflecting at the top and bottom surfaces of the substrate. The measurements here are taken with an illumination spot diameter of 30 μm. This confirms that both the finite array size and varying incident angles have a negligible effect on the optical response of the surface. Notably, the measured quality factors are higher than expected from simulations (see Fig. 1). However, when fabrication imperfections, such as non-vertical sidewalls and a structure undercut, are accounted for in the simulation, the calculated quality factors increase and good agreement with the measured transmission spectrum is obtained (see Supplementary Fig. 2). The measured quality factors of our uniformly-sized metasurfaces are higher or on par with measured quality factors of metasurfaces based on asymmetry-induced quasi-BIC[12,15,41], toroidal modes[45] or electromagnetically induced transparency[46]. However, unlike these other designs, our metasurfaces allow for local manipulation of the resonance phase (see Supplementary Note 4). The measured near-complete extinction of the transmission on resonance is evidence that the nanoblocks are fabricated with a high degree of size uniformity. A comparison with electromagnetic simulations suggests the nanoblocks are fabricated with a standard deviation in the length $L$ of less than 6 Å (see Supplementary Note 5). For silicon, this represents a near single atomic layer precision in nanostructuring of the metasurface and a significant improvement over previous processes for metasurface fabrication with standard deviations of 2.1 nm[47].

To selectively deflect light over a narrow wavelength range, we imprinted a linear phase gradient on the transmitted light by varying the nanoblock side lengths along one of the metasurface in-plane directions (see Supplementary Table 1 for metasurface design parameters). The nanoblock size lengths are set based on the phase look-up table determined from numerical simulations in Fig. 1c by applying a traditional forward design. We performed Fourier plane imaging of the transmitted light to characterize the light deflection of the metasurface. Figure 3a shows the measured spectral diffraction efficiency and images of the Fourier plane of the light transmitted through a high-Q beam deflector metasurface in the on- and off-resonance case for deflecting $x$-polarized light along the $y$ direction. This results in a TE polarization of the deflected light, referred to as TE deflection. On resonance, light is preferentially deflected to an angle of $\theta = 26°$ ($\varphi = 0°$) from the surface normal as determined by the linear phase gradient of the metasurface along the $y$ direction corresponding to $2\pi/4P$. At the design wavelength $\lambda_R = 1293$ nm, a diffraction efficiency of 41.2% is attained. The remaining power is coupled to the normal and opposite direction at $\theta = -26°$. Notably, in a representative off-resonant case, at $\lambda_O = 1288$ nm, 97.4% of the transmitted light remains in the surface normal direction. The measured quality factor of the TE light deflection is $Q = 1191$. By imprinting a phase gradient along the orthogonal direction (i.e., along the $x$ direction) of the metasurface, light of the same polarization is deflected along the $x$-direction to an angle $\varphi = 26°$ ($\theta = 0°$) with a maximum diffraction efficiency of 17.1% at $\lambda_R = 1293$ nm (see Fig. 3b), due to the polarization-independent response. The result is a TM polarization of the deflected light, referred to as TM deflection. Here, a larger quality factor of $Q = 1458$ is measured. Off-resonance, at $\lambda_O = 1288$ nm, 98.1% of the transmitted light remains in the normal direction. Two additional representative measurements of TE and TM deflection are illustrated in Supplementary Figs. 3 and 4.

The operating wavelength for beam deflection can be adjusted by shifting the average length of nanoblocks. Figure 3c, d illustrates the measured spectral diffraction efficiency and quality factors for TE and TM deflection for surfaces with varying operating wavelength and fixed phase gradient. By modifying the phase gradient imprinted on the surface, light can be deflected to different angles. Figure 3e, f illustrates the measured spectral diffraction efficiency for TE light deflection and quality factors for TE and TM deflection for surfaces

deflecting light to different angles (the corresponding TM diffraction efficiency is shown in Supplementary Fig. 5). Overall, the obtained diffraction efficiencies on resonance for the desired deflection angle range between 22–76.6% and 4–19.1% for the TE and TM deflection, respectively. The lower diffraction efficiency obtained for TM polarized deflection compared to TE is likely due to nearest-neighbor coupling between the nanoblocks. For TM deflection, the nearest-neighbor coupling occurs along the direction of the phase gradient, therefore making this case more prone to local phase errors that arise from coupling between nanoblocks and fabrication imperfections. The attained $Q$ factors for TE and TM deflection range between $Q_{TE} = 298$–1191 and $Q_{TM} = 288$–1475. These results illustrate that light can be deflected along two dimensions to arbitrary angles $\theta$ and $\varphi$ with high quality factors. This contrasts with previous demonstrations of high-quality factor metasurfaces based on guided mode resonance structures, which are inherently one-dimensional in their light deflection capability, and require large illumination apertures and precise configuration of the incidence angle[30,31].

To demonstrate the wavefront shaping capabilities of our metasurfaces, we realize high-quality factor radial metalenses that focus light along two dimensions over a narrow wavelength range. The metalenses are designed by imposing a paraboloidal phase profile on the transmitted light by setting the nanoblock size lengths according to the look-up table in Fig. 1c. Figure 4a, b illustrates the measured electric field intensity in the focal plane and in a cross-section along the optical axis of a metalens with a numerical aperture (NA) of 0.1 at the resonance wavelength of $\lambda = 1276$ nm. On resonance, light is symmetrically focused to a near-diffraction-limited focal spot. However, at $\lambda = 1266$ nm, at a wavelength only 10 nm away from the resonance wavelength, light propagates through the metalens without focusing (see Fig. 4c, d). Figure 4e, shows the measured spectral focusing efficiency of the metalens, highlighting its wavelength-selective

operation. The maximum focusing efficiency is 25.3% on resonance. From the spectral focusing efficiency, a quality factor of $Q = 531$ is determined for the metalens. The measured transmittance of the lens on resonance is 55.6% (see Supplementary Fig. 6). Figure 4f illustrates measured quality factors from different lenses with numerical apertures ranging from 0.1 to 0.8 and varying operating wavelengths. A similar trend in increased quality factors towards shorter wavelengths, as in Fig. 2b, is observed. The highest quality factor $Q = 880$ is obtained for a lens with 0.18 NA (see Supplementary Fig. 7). To our knowledge, this represents the highest quality factor radial lens demonstrated to date and is one order of magnitude higher than previous demonstrations with nonlocal metasurfaces[34]. Figure 4g–j illustrates the measured electric field intensity in the focal plane for metalenses with numerical apertures of 0.18, 0.4, 0.6, and 0.8. The observed full width at half maximum of the focus is in good agreement with the diffraction-limited values. Additional representative measurements of metalenses are shown in Supplementary Fig. 7. The Strehl ratios of the characterized lenses range between 0.1 and 0.57 (see Supplementary Fig. 8). Both the measured Strehl ratios and focusing efficiencies decrease with increasing numerical aperture, which we attribute to an increased wavefront distortion due to coupling between neighboring elements, which increases with larger phase gradients. We note that the performance of the metalenses demonstrated here is currently limited by fabrication accuracy. Furthermore, the coupling between neighboring elements is not optimized in the design but is known to have a significant effect[43].

In wavefront shaping with high quality factors, there is an inherent tradeoff between quality factors and accurate phase sampling due to fabrication limitations. This is due to the rapid variation of the phase on resonance. For example, in the structure reported here, the phase of the transmitted light varies with 9.4 rad/nm within the range of $\pi/2$ to $3\pi/2$ (see Fig. 1b). As a result, a precision of the nanorod side length

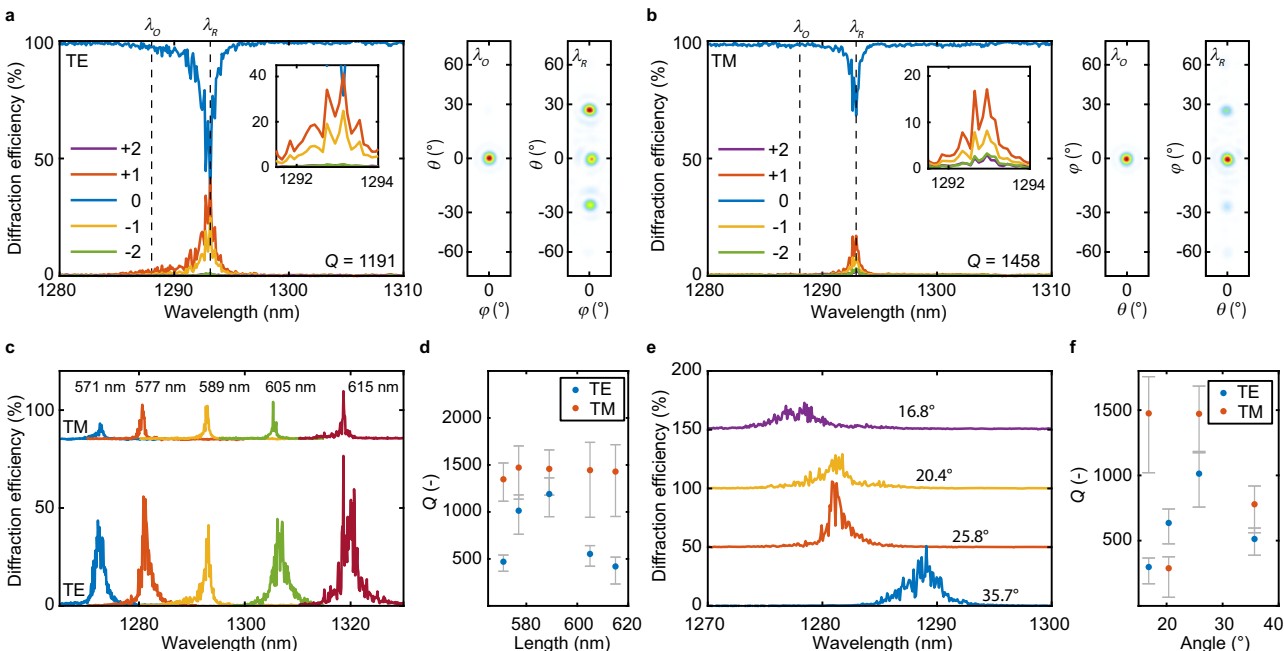

**Fig. 3 | High-quality factor beam deflection along two dimensions.**
**a, b** Experimentally measured diffraction efficiencies of the −2 (green), −1 (yellow), 0 (blue), +1 (red), and +2 (purple) diffraction orders and Fourier plane images of a metasurface showing **a** TE deflection of x-polarized light along the y direction and **b** TM deflection of x-polarized light along the x direction. The desired diffraction order is +1, with $\theta = 26°$ and $\varphi = 26°$, respectively. The insets show a zoomed-in region of the plots. **c** Measured diffraction efficiency of TE and TM light deflection in the +1 diffraction order with varying average nanoblock side lengths. The values

for TM deflection are shifted vertically by 85% for better illustration. **d** Measured quality factors of TE (blue circles) and TM (red circles) light deflection with varying average nanoblock side lengths. **e** Measured diffraction efficiency of TE light deflection with varying deflection angle. The curves are shifted vertically by 50% from each other for better visibility. **f** Measured quality factors of TE (blue circles) and TM (red circles) light deflection with varying deflection angles. The error bars in (**d**) and (**f**) illustrate the 95% confidence interval of the fit. For all metasurfaces $P = 736$ nm and $H = 695$ nm. The metasurface size is 150 μm × 150 μm.

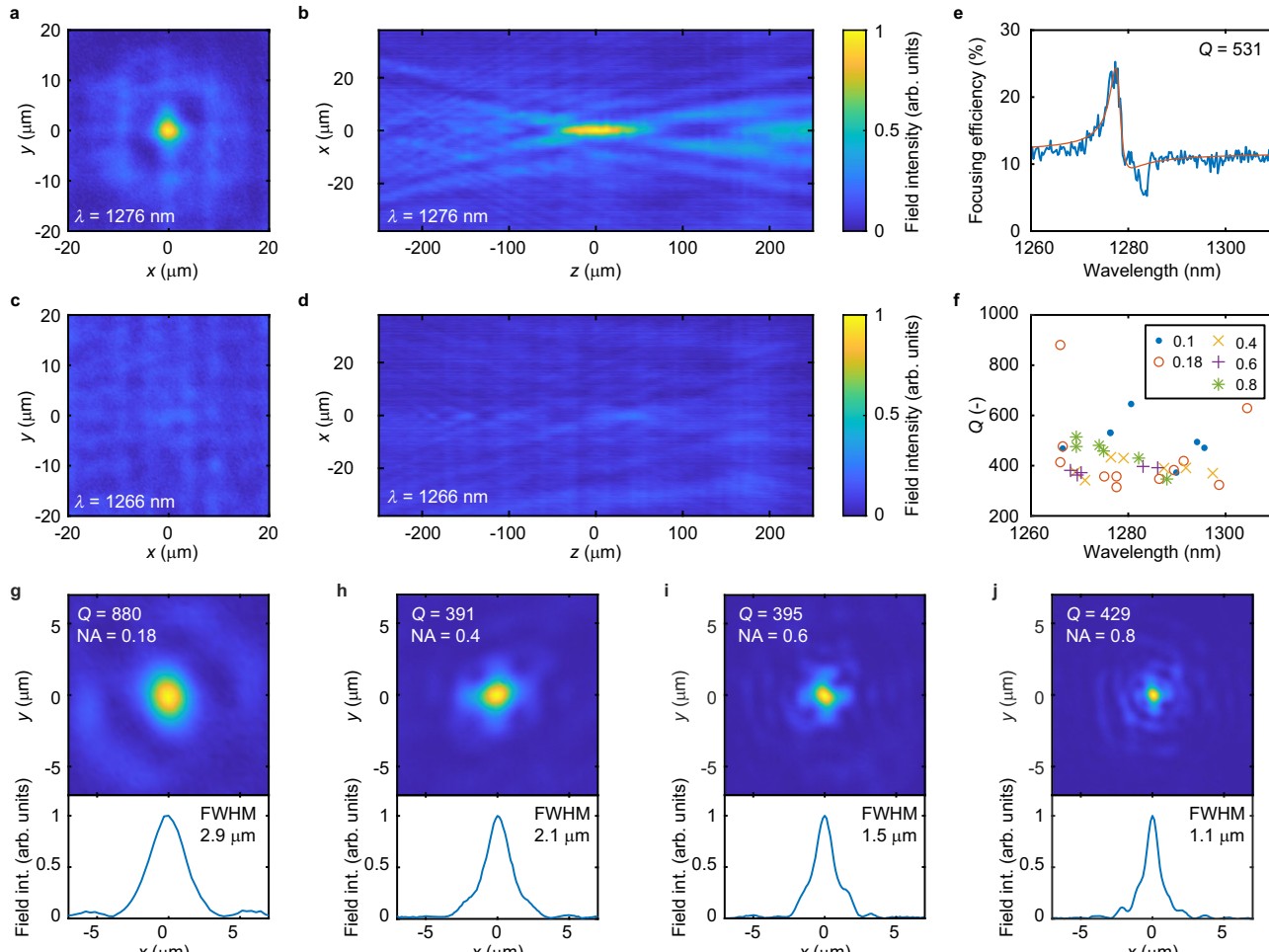

**Fig. 4 | High-quality factor radial metalenses for focusing along two dimensions. a** Measured field intensity at the focal plane (*x*−*y* plane) on resonance at $\lambda = 1276$ nm of a high-Q metalens with 0.1 NA. **b** Measured field intensity along the optical axis in the *x*−*z* plane on resonance. **c** Measured field intensity at the focal plane off-resonance at a representative wavelength $\lambda = 1266$ nm. **d** Measured field intensity along the optical axis in the *x*−*z* plane off-resonance. The scaling of the color maps in (**a**–**d**) is identical. **e** Measured spectral focusing efficiency of the metalens with 0.1 NA and Q = 531. The red line shows a Fano fit to the measurement.

**f** Measured quality factors of fabricated lenses with different resonance wavelengths and numerical apertures of 0.1 (blue disk), 0.18 (red circle), 0.4 (yellow cross), 0.6 (purple plus), and 0.8 (green asterisk). **g**–**j** Measured field intensity (field int.) at the focal plane (*x*−*y* plane) on the resonance of four high-Q metalenses with numerical apertures of 0.18, 0.4, 0.6, and 0.8, respectively. The resonance wavelengths are 1266 nm, 1291.8 nm, 1285.5 nm, and 1281.4 nm, respectively. The insets denote the full width at half maximum (FWHM) of the focusing. The metalenses are 100 μm in diameter P = 736 nm and H = 695 nm.

of 0.17 nm is required to sample the phase at a $\pi/2$ increment in this range. Furthermore, this required precision generally becomes stricter when increasing the quality factor. Our fabrication process results in an accuracy of the nanoblock side length of less than 0.6 nm, which produces considerable errors in phase sampling, in turn causing the appearance of stray light in the focal plane of the fabricated metalenses (Fig. 4j). This shows that fabrication imperfections are currently a limiting factor in the demonstrated quality factors and diffraction efficiencies. We expect an improved performance with higher fabrication uniformity and lower side wall surface roughness. While there is still room for improving the electron beam lithography-based process implemented here, using methods such as scanning probe lithography or atomic layer etching may allow fabrication with near-atomic layer accuracy. Another approach is to employ different optical modes and unit-cell geometries that enable wavefront shaping with similar quality factors but with more relaxed fabrication requirements. For example, a configuration in reflection with similar quality factors shows a slower variation of the phase of 2.64 rad/nm requiring only a 0.6 nm precision to sample the phase at a $\pi/2$ increment in the $\pi/2$ to $3\pi/2$ range (see Supplementary Note 6). Furthermore, accounting for fabrication imperfections in the device design is also expected to result in better

device performance. This could be done by creating a new iteration of the phase look-up table (Fig. 1c) by considering the fabrication imperfections identified in Supplementary Fig. 2. In addition to fabrication constraints, the local approximation used to design the phase profile along the array also limits the maximum efficiency of the devices due to non-negligible long-range coupling between unit cells. However, this is not a fundamental limitation, and improved performance can be attained by accounting for nanoblock coupling in the design[48], applying design optimization approaches or by employing inverse design concepts[49]. Recently, topological optimization has been employed to realize high-Q metagratings for 1D beam deflection[50]. Relying on similar concepts, we demonstrate that particle swarm optimization can be used to optimize the design of our structure for TM light deflection resulting in an increase in the diffraction efficiency from 47% to 82% (see Supplementary Fig. 9).

In summary, we have demonstrated the concept of a higher-order Mie-resonant metasurface as a pathway towards high-quality factor two-dimensional wavefront manipulation. By spectrally overlapping an electric dipole and electric octupole mode, a sharp resonance in the transmission is obtained that enables local control of the wavefront of light. Using local phase control, we realize beam deflectors with high

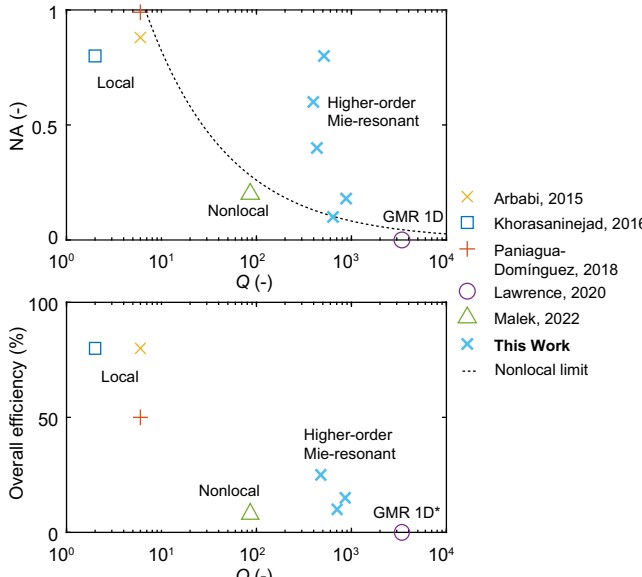

**Fig. 5 | Comparison of reported metasurfaces for two-dimensional wavefront manipulation.** Comparison of our results (light blue crosses) to previously reported experimental values[23,24,30,34,54] for two-dimensional metalenses in terms of quality factor, numerical aperture, and overall efficiency from Arbabi et al.[23] (yellow cross), Khorasaninejad et al.[24] (blue square), Paniagua-Domínguez et al.[54] (red plus) and Malek et al.[34] (green triangles). The dotted black line shows the upper bound on the numerical aperture for nonlocal metasurfaces[33,34]. The results for guided mode resonance (GMR) are shown for comparison[30] (purple circle), but only one-dimensional wavefront manipulation is possible. *No overall efficiency is reported for GMR[30].

directivity and quality factors of $Q = 288-1475$. We further demonstrate radial lensing with near-diffraction-limited focusing and quality factors of $Q = 314-880$. Notably, due to the local nature of these modes, high quality factors are observed even with small illumination areas, here 30 μm in diameter. This is in contrast to recent work with guided-mode resonance structures, where illumination spots of up to 500 μm are required to attain considerable quality factors[30]. Although the geometric tolerances of the proposed scheme are demanding, we demonstrate a robust and consistent fabrication of the devices. The measurements presented here were taken from five different samples fabricated with high yield and high reproducibility. The demonstrated quality factors are currently limited by fabrication imperfections and coupling between neighboring structures. Higher quality factors may be realized by using structures supporting different higher-order Mie modes. The amplitude variation across the resonant mode can be reduced by modifying the device design, for example, by operating in reflection (see Supplementary Note 6) or by using other types of higher-order modes such as magnetic octupole, hexapole, or toroidal modes.

The high level of wavefront control and strong light interactions with our structure, as evidenced by the high quality factor, make our higher-order Mie-resonant metasurfaces highly suitable for optical sensing[15], nonlinear optics[51], directional lasing[52] and active wavefront manipulation[53]. Figure 5 illustrates a comparison of the state of the art of two-dimensional wavefront manipulation in terms of experimentally demonstrated quality factor, numerical aperture, and overall efficiency for radial metalenses[23,24,30,34,54]. The concept reported here enables wavefront control with unprecedented quality factors and high numerical aperture. As compared to nonlocal metasurfaces[33,34], our method is polarization independent, robust against incidence angle variations and shows higher quality factor, efficiency, and numerical aperture (see Supplementary Note 7 for a detailed comparison to previous work). Furthermore, in nonlocal metasurfaces the

quality factor for a given numerical aperture is limited by the band-structure, and a single-layer surface shows an upper bound of 25% efficiency[33,34]. The method demonstrated here does not show any of these limitations. Our design requires no additional polarization optics and is also advantageous for the implementation of active optical devices that dynamically modulate the dielectric environment of a metasurface unit cell. For example, by introducing a refractive index change in the nanoblocks reported here using either the thermo-optic or electro-optic effect, our metasurface may be used to dynamically steer light[55]. The high-$Q$ metasurfaces demonstrated here may also be realized at visible wavelengths, where the narrow spectral response is advantageous for display and coloring applications[56,57]. We envision that the extraordinary characteristics of higher-order Mie resonances in high-index nanoblock arrays demonstrated here will lead to numerous applications, as well as new physics[58].

## Methods
### Experiment
The fabricated metasurfaces were characterized on a home-built optical transmission microscope (schematically illustrated in Supplementary Fig. 10). Coherent light from a wavelength-tunable diode laser (Santec TSL-510) was loosely focused on the metasurface. The transmitted light was collected with an objective lens (20×, 0.4 NA, Mitutoyo) and projected either onto an InGaAs IR camera (Xenics Bobcat 320) or a power meter (Thorlabs S122C). For measuring the transmission spectrum, the laser wavelength was scanned, and the transmitted power was recorded with the power meter. For the power normalization, the sample was removed, and the illuminated power was recorded through the same area. For the beam deflection measurements, a Fourier plane was imaged onto the camera and a 0.9 NA objective lens was used to capture all diffraction orders. For characterizing the lenses, the focal plane was imaged onto the camera and a scan along the optical axis was obtained by moving the surface along the z direction.

The transmission of the metasurfaces, $T = P_T/P_I$, is calculated by recording the power transmitted through the metasurface, $P_T$, normalized by the power incident on the metasurface, $P_I$. For the beam deflectors, the diffraction efficiency is defined as the fraction of transmitted power coupled into a specific diffraction order. To this end, the intensity around each diffraction order is integrated within a square with a side length of 4 FWHM of the intensity of the diffraction order. For the metalenses, the focusing efficiency is defined as the fraction of transmitted power coupled into a circle around the focal spot with a radius of 2 times the airy disk radius.

Scanning electron micrographs were acquired on an FEI Nova 200 NanoLab system to measure the sizes of the fabricated structures. For imaging, the surfaces were covered with a 2 nm thick gold layer by sputter deposition.

### Fabrication
The metasurfaces were fabricated on borosilicate glass substrates ($n = 1.503$) with a thickness of 220 μm. To remove organic residues from the surface, the substrates were cleaned in an ultrasonic bath in acetone, isopropyl alcohol, and deionized water each for 15 min, dried using an $N_2$ gun, and subsequently cleaned using oxygen plasma. Amorphous silicon was deposited onto the glass using plasma-enhanced chemical vapor deposition. In a subsequent step, the nanoblocks were written in a spin-coated MaN−2403 resist layer by standard electron beam lithography. The nanoblocks were then transferred to the amorphous silicon using a $SiO_2$ hard mask with chlorine-based inductively coupled reactive ion etching. As a last step, the residual mask was removed by immersing the samples in buffered hydrofluoric acid (1:7) for 5 s and subsequent rinsing in deionized water. The uniform and beam deflector metasurfaces were fabricated on an area of 150 μm × 150 μm. The metalenses were fabricated with a

diameter of 100 μm and a parabolic phase profile according to the equation

$$\varphi(x,y) = \frac{2\pi}{\lambda}\left(\sqrt{x^2+y^2+f^2}-f\right), \qquad (1)$$

where $\lambda$ is the design wavelength and $f$ the focal length. All metalenses were designed for a wavelength of $\lambda = 1280$ nm. The variation of the nanorod side length is set according to Fig. 1c with a discretization of 0.1 nm. Further metalens design parameters are given in Supplementary Table 2.

## Simulation

The numerical modeling of the nanostructures was carried out using an FDTD method. Simulations were performed with commercially available FDTD software (Lumerical FDTD Solutions). A constant refractive index of $n = 1.503$ was used for the borosilicate glass, and a constant value of $n = 3.45$ for amorphous silicon, as experimentally determined by ellipsometry. The simulations were carried out with a spatially coherent plane wave illumination and periodic boundary conditions were applied on all sides of the computational domain unless otherwise noted. The smallest mesh-refinement of 5 nm was used. For the oblique illumination simulations, the broadband fixed angle technique is used.

## Data availability

All the relevant data of this study are included in the paper and supplementary information file, and raw data are available from the corresponding author upon request.

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

## Acknowledgements

This work was supported by the Air Force Office of Scientific Research under grant FA9550-18-1-0354 (C.U.H., R.S., and H.A.A.) and the Meta-Imaging MURI grant #FA9550–21–1-0312(M.F.). C.U.H. also acknowledges support from the Swiss National Science Foundation through the Early Postdoc Mobility Fellowship grant #P2EZP2_191880. L.M. acknowledges support from the Fulbright Fellowship program and the Breakthrough Foundation. We gratefully acknowledge the critical support and infrastructure provided for this work by The Kavli Nanoscience Institute at Caltech.

## Author contributions

C.U.H, H.A.A., and R.S. conceived the project. C.U.H. performed the simulations, fabricated the devices, built the experiment, performed the measurements, and analyzed the results. R.S. and C.U.H. conceived the metasurface design. M.F. assisted with simulations and fabrication. L. M. assisted with analyzing the results. C.U.H. wrote the paper with input from all other authors. H.A.A supervised all aspects of the project.

## Competing interests

The Authors declare no competing interests.
