## [Peer Review File · Nature Communications]

REVIEWER COMMENTS

Reviewer #1 (Remarks to the Author):

In the response letter, the authors clarify the advantages of this work, particularly the use of higher-order Mie-resonant modes. This aspect is important as it could potentially break the 25% efficiency limit and allows for high-NA wavefront control. The paper also emphasizes other superior properties such as reduced spectral shift, less angular dependence and localized modes. We believe that with these clarifications, the manuscript is more suitable for Nat Comm.

However, further clarification should be made on the following points. First, as pointed out by other reviewers, the metasurface design still relies on a conventional phase array based on local approximation, which isn't optimal for high-Q metasurfaces that exhibit long-range coupling (the localized high-order modes here still expand several unit cells). This will fundamentally limit the efficiency of the devices, particularly for high NA. Second, as mentioned in the authors' response letter, "symmetry-protected Bound States in the Continuum (BIC) structures cannot be used for beam steering as their mode is delocalized in the array." This is only the case for conventional design approaches. On the other hand, the authors seem to have overlooked a recent study that employs multi-resonant nonlocal metasurfaces (Zhou, You, et al. "Multiresonant Nonlocal Metasurfaces." *Nano Letters* (2023)). This work applies topology to account for long-range interactions in symmetry-protected BIC, managing to achieve high efficiency for an NA greater than 1 (in an $n=1.45$ media). A comparative analysis with this work would be valuable.

Reviewer #2 (Remarks to the Author):

I have previously reviewed this manuscript for another journal. The authors have partially responded to my comments, but not completely. In particular, the authors did not address my main concern.

In my previous report, my main comment concerned the sampling of the phase, which is a crucial issue for wavefront shaping applications: "It is impossible to realize a correct sampling of the phase between 0 and 2π with high-quality factor meta-atoms". In their response, the authors put forward the quality, accuracy, and robustness of the fabrication, but they do not discuss the issue of phase sampling.

Table 1 and Table 2 added in the revised Supplementary Information are a good illustration of my comment about the intrinsic limitations of phase sampling with high-quality factor meta-atoms.

According to Supplementary Table 1, the lengths of the central nanoblocks chosen to sample the $[0, 2\pi]$ interval should be incremented by only 0.2 nm. This very small length increment is a direct consequence of the high quality factor. However, the authors have estimated that the accuracy of their fabrication process is 0.6 nm (standard deviation of a gaussian distribution), 3 times larger than the desired length increment.

These data show that the sizes of the central nanoblocks cannot be controlled with a sufficient accuracy, which should be of 0.1 nm or smaller to be sure that the length uncertainty remains smaller than the theoretical length increment between adjacent nanoblocks. If this condition is not fulfilled, the phase values realized on the sample can be largely different from the desired ones, since the phase variation with the geometry is very steep. Moreover, there is a high risk to realize a non-monotonous and chaotic phase variation instead of a monotonous variation. In other words, the phase cannot be correctly sampled because of the high quality factor.

The problem is even worse for the lenses: since the central Fresnel zones are composed of a few tens of nanoblocks (see Supplementary Table 2), the theoretical length increment between adjacent nanoblocks is probably smaller than 0.2 nm.

Note that a tiny variation of any other geometrical parameter (sidewall tilt, undercut...) will have a similar detrimental and uncontrolled impact on the phase sampling.

This issue of phase sampling is an intrinsic limitation of wavefront shaping with high-quality factor meta-atoms. Even if the authors demonstrate an excellent control over the fabrication, it is not sufficient to perform wavefront shaping with good performance. There is a trade-off between the quality factor of the meta-atoms and the performance of wavefront shaping. In my opinion, this crucial point should not be underestimated. It is misleading for potential readers to promote wavefront control with high quality factors without discussing the intrinsic limitations of the approach.

The incorrect phase sampling is probably the main reason for the small efficiencies reported by the authors (<50% for the beam deflectors and 20% for the lens). More importantly, an incorrect phase sampling results in the presence of a lot of stray light. For the beam deflectors, undesired diffraction orders carry a large fraction of the transmitted power at resonance. In TE polarization, the zeroth order diffraction efficiency is on the order of 50%, a value that is roughly equal to the diffraction efficiency of the useful first order, see Fig. 1a. In TM polarization, it is worse since the zeroth order carries more power than the first order, see Fig. 1b. These values show that the phase produced by the metasurface is significantly different from the desired one. Figure 4 also shows a lot of stray light around the focal spot of the different lenses.

I do not recommend publication of this manuscript in Nature Communications.

Additional comments:

- In the revised version of the Supplementary Information, the authors have added the measurements of the transmission spectrum of the lenses (Sup Fig 6). The authors should define what they mean by “transmission” and “diffraction efficiency” (Fig. 4e and Sup Fig 7). In particular, the difference between the two quantities should be clearly indicated. Concerning the measurement of the diffraction efficiency, the authors should explain how they separate the different diffraction orders.

- The authors have added in the Supplementary Information important details about the realization of the deflectors and lenses (Sup Tables 1 and 2). Given its importance, the information about the size of the devices (150 μm for the deflectors and 100 μm for the lenses) should be included in the main text. The fact that the lens design is based on a parabolic phase profile should also be included in the main text.

- The calculation of the Strehl ratio of the lenses is done by integrating the intensity in the focal plane around the focal spot within a radius of 4 times the diffraction-limited Airy disk radius. Since the intensity spreads in the focal plane beyond 4 times the Airy disk radius (according to Fig. 4 and Sup. Fig. 7), a non-negligible fraction of light may be missed. The calculations of the Strehl ratio should be done by integrating the intensity in the focal plane within a larger radius.

Reviewer #3 (Remarks to the Author):

In this paper the authors have used localized high-Q resonances in Mie-metasurfaces for linear optical components that can work over a narrow wavelength range. Although most of the community aims to achieve high performance optical functionality over a wide wavelength range, there are cases where high performance optical meta-optics over a narrow bandwidth could be beneficial. The fabrication of these metasurfaces is impressive, the results are good and the work shown in the main manuscript and its supplemental is very thorough. There are several issues though that range from fundamental to minor, and they are described below:

1. To set the record straight, achieving high-Q resonances using interference of two modes was pioneered by A. Kodigala, T. Lepetit, Q. Gu, B. Bahari, Y.F. Nature, and 2017, “Lasing action from photonic bound states in continuum,” Nature 541, 196–199 (2017) in the context of photonic crystals. Later, Kivshar et al. used this concept with Mie-metasurfaces and termed it super-cavity modes in a number of papers (see for example: M.V. Rybin, K.L. Koshelev, Z.F. Sadrieva, K.B. Samusev, A.A. Bogdanov, M.F. Limonov, and Y.S. Kivshar, “High-Q Supercavity Modes in Subwavelength Dielectric

Resonators,” Phys Rev Lett 119(24), 243901 (2017), K. Koshelev, S. Kruk, E. Melik-Gaykazyan, J.-H. Choi, A. Bogdanov, H.-G. Park, and Y. Kivshar, “Subwavelength dielectric resonators for nonlinear nanophotonics,” Science 367(6475), 288–292 (2020)). Please note that although Kivshar et al. termed these modes also BICs or quasi-BICs, the physics is exactly the same as what this papers uses, i.e., using interference of two modes in individual resonators rather than non-local modes. Even the concept of placing these resonators on top of a conducting surface was already mentioned in these papers.

None of this prior work has been cited in the paper which leads to the wrong impression that the authors have invented a new, non-local way of achieving high-Q states/resonances that can be used for optical components. Granted, most of the prior literature on “super-cavity modes” has dealt with lasing or nonlinear optical phenomena, but the physics is the same and the concept can be applied to linear optical phenomena too. Maybe the authors of this manuscript missed this prior work because it was always portrayed using a mixed language that included “BIC modes”. I think that this is a glaring omission and the claims of novelty should be scaled accordingly.

2. As prior reviewers have noted, the authors don’t provide a clear path on how to design these high-Q resonances, other than numerical optimization. The lowest Mie modes of the resonators can be approximately calculated using analytical formulas, but the higher order modes are very sensitive to the detailed geometry of the structures. Moreover, the phase gradient numbers quoted in table S1 and used in the design of the metaoptics have to depend on the Q of the spectral features. But there is no way to know a-priori what this Q will be since this value will depend on a myriad of factors. A big part of scientific publications has to do with reproducibility, so how exactly could another group replicate these results? Does one first have to go through a fabrication step to figure out the Q’s and parameter sensitivity, and only then one can design a phase profile using these resonances?

3. Angular dispersion: the authors show very nice results of low angular dispersion which in turn make the design of metaoptics possible. However, there is a claim in several parts of the paper that this is a feature of Mie resonant metasurfaces. I would like to point out that this is not a universal statement. When the meta-atom has a design that is close to isotropic (for example, close to cube or sphere), then the modes excited by plane waves at different angles of incidence will show low angular dispersion. However, the modes of a metaatom that deviates from this 1:1 aspect ratio will show a very large angular dispersion (think for example of a flat cylinder or prism as an extreme case)..

4. If the claim that the spectral feature used in the paper is due to interference of the radiation pattern of two modes of an individual resonator, then I don’t understand why the authors don’t just show that in a simulation, instead of dedicating an entire section of the supplemental to mode volume, studying the spectral changes when one perturbs one resonator in an array, etc. Why not just perform a simulation with one resonator with absorbing boundaries and no periodic boundary conditions?

5. some of the choice of language is a bit problematic. Why do the authors choose to use “higher-order Mie resonant metasurface”? All this paper is showing is how to use the interference of two Mie modes for optical components. Same for the use of “ED/EO mode” (or maybe use “supercavity mode” as it was used in the past).

6. why does the Q-factor increase when there is nearest neighbor coupling? Usually coupling leads to lowering of the Q.

7. Minor issue: several of the graphs are very small and have traces with overlapping lines & colors that can't be discerned.

In summary, I think that this paper shows very nice and impressive results but the novelty claim should be recalibrated due to the glaring omission of prior work that was used in the context of other optical phenomena. The issues that I raised are serious enough that warrant a new version of the manuscript.

Response to Reviewer Comments

In this document the reviewer comments are pasted in blue, our response in black and the parts newly added to the manuscript in red.

Reviewer #1:

In the response letter, the authors clarifies the advantages of this work, particularly the use of higher-order Mie-resonant modes. This aspect is important as it could potentially break the 25% efficiency limit and allows for high-NA wavefront control. The paper also emphasize other superior properties such as reduced spectral shift, less angular dependence and localized modes. We believe that with these clarifications, the manuscript is more suitable for Nat Comm.

We thank the reviewer for pointing to some of the merits of our work, and for highlighting the suitability for publication in Nature Communications.

However, further clarification should be made on the following points.

1. **Comment:** First, as pointed out by other reviewers, the metasurface design still relies on a conventional phase array based on local approximation, which isn't optimal for high-Q metasurfaces that exhibit long-range coupling (the localized high-order modes here still expands several unit cells). This will fundamentally limit the efficiency of the devices, particularly for high NA.

Response: We agree with reviewer that a local approximation limits the efficiency of the devices, particularly for high NA. In general, the best performance for metasurfaces will be obtained by performing an optimization over the entire metasurface aperture. In most cases such optimization is currently computationally intractable, especially for large array sizes, due to the large simulation area required which dictates the resulting large computational cost. We envision that using the current design based on the local approximation as a starting point for using optimization methods to obtain even higher values of quality factor, numerical aperture and efficiency as illustrated in Fig. 5.

To address this comment, we expanded the discussion on optimization approaches to improve the metasurface performance on page 14 of the revised manuscript as follows:

In addition to fabrication constraints, the local approximation used to design the phase profile along the array also limits the maximum efficiency of the devices due to non-negligible long-range coupling between unit cells. However, this is not a fundamental limitation, and an improved performance can be attained by accounting for nanoblock coupling in the design⁴⁸, applying design optimization approaches or by employing inverse design concepts⁴⁹. Recently, topological optimization has been employed to realize high-Q metagratings for 1D beam deflection⁵⁰. Relying on similar concepts, we demonstrate that a particle swarm optimization can be used to optimize the design of our structure for TM light deflection resulting an increase in the diffraction efficiency from 47% to 82% (see Supplementary Fig. 9).

Additionally, we also performed a numerical design optimization of the TM beam deflection, to illustrate how design optimization can lead to improved efficiencies. The result of this optimization is illustrated in the newly added **Supplementary Fig. 9**.

Supplementary Figure 9 | Numerical optimization of high-quality factor TM beam deflection to $\varphi = 35.8^\circ$. **a, b,** Simulated diffraction efficiencies of the -1, 0, and +1 diffraction orders and Fourier plane images of a metasurface showing TM deflection of x-polarized light along the x direction for (a) a forward design structure and (b) an optimized structure using a particle swarm optimization. The desired diffraction order is +1, with $\varphi = 35.8^\circ$. On resonance, $\lambda_R = 1291.9$ nm, a diffraction efficiency of 46.5% and 81.9% is attained for the forward design and the optimized design, respectively. The design includes 3 nanoblocks per Fresnel zone and nanoblock side lengths are [553.9, 554.9, 557] nm and [554.8, 554.9, 558] nm for the forward design and optimized design respectively. For the optimization the rod lengths L_1 and L_3 are varied and L_2 is fixed. Additionally, the smallest mesh refinement is set to 10 nm, to reduce the computational cost of the optimization.

- Comment:** Second, as mentioned in the authors' response letter, "symmetry-protected Bound States in the Continuum (BIC) structures cannot be used for beam steering as their mode is delocalized in the array." This is only the case for conventional design approaches. On the other hand, the authors seem to have overlooked a recent study that employs multi-resonant nonlocal metasurfaces (Zhou, You, et al. "Multiresonant Nonlocal Metasurfaces." Nano Letters (2023)). This work applies topology to account for long-range interactions in symmetry protected BIC, managing to achieve high efficiency for an NA greater than 1 (in an $n=1.45$ media). A comparative analysis with this work would be valuable.

Response: We thank the reviewer for pointing out this relevant and very recent publication that we had overlooked in the resubmission of the manuscript. We have added the following sentence with a reference to this work to the revised manuscript on page 14:

Recently, topological optimization has been employed to realize high-Q metagratings for 1D beam deflection⁵⁰.

Additionally, we added a note to Supplementary Note 4 on page 18 of the revised Supplementary Information, referring to this interesting recent development.

While conventional design approaches fail to achieve wavefront manipulation, notably, topological optimization can be employed to realize high-Q metagratings for 1D beam deflection with symmetry protected q-BIC structures¹¹.

Finally, as suggested by the reviewer, we included a **newly added comparative analysis**, together with our comparison to previous work in Supplementary Note 7 including this work and also other works on high-Q 1D beam deflection.

Finally, we compare our work to the previous work on one dimensional beam deflection with high quality factors. Supplementary Table 4 illustrates a comparison of key performance metrics of the state of the art of one dimensional high-quality factor beam deflection^{11,20,21}.

	Method	Q	Diffraction efficiency	Polarization dependence	Maximum deflection angle	Wavefront manipulation	Angular dispersion
Ref. 11	q-BIC	~25	35%*	Yes	43° (46° in glass, simulation)	1D	Not reported
Ref. 20	GMR	2500	15%	Yes	42° in air	1D	Large
Ref. 21	GMR	~380	56.7%	Yes	32° in air	1D	Large
This work	Higher order Mie resonances	1458	55.9%	No	35.4° in air	2D	Small

Supplementary Table 4 | Comparison to the state of the art of high-Q one dimensional wavefront manipulation. Comparison of the current state of the art of one-dimensional beam deflection with high quality factor considering key performance metrics for metasurface optical elements. Experimentally reported values are compared due to the crucial importance of fabrication imperfections when realizing these devices. The quality factor here represents the quality factor of 1D light deflection. Quality factors for Ref. 11 and 21 were approximated from figures. *As opposed to the other values of the diffraction efficiency, the value of Ref. 11 is normalized to the total incident light rather than total transmitted light. Experimental transmission is not reported in Ref. 11.

In closing, we thank the reviewer for their valuable input regarding design optimization and a relevant study that we had overlooked. We believe that in addressing their comments we have significantly improved the quality of the manuscript and its suitability for publication in Nature Communications.

Reviewer #2:

I have previously reviewed this manuscript for another journal. The authors have partially responded to my comments, but not completely. In particular, the authors did not address my main concern.

1. **Comment:** In my previous report, my main comment concerned the sampling of the phase, which is a crucial issue for wavefront shaping applications: “It is impossible to realize a correct sampling of the phase between 0 and 2π with high-quality factor meta-atoms”. In their response, the authors put forward the quality, accuracy, and robustness of the fabrication, but they do not discuss the issue of phase sampling.

Table 1 and Table 2 added in the revised Supplementary Information are a good illustration of my comment about the intrinsic limitations of phase sampling with high-quality factor meta-atoms. According to Supplementary Table 1, the lengths of the central nanoblocks chosen to sample the $[0,2\pi]$ interval should be incremented by only 0.2 nm. This very small length increment is a direct consequence of the high quality factor. However, the authors have estimated that the accuracy of their fabrication process is 0.6 nm (standard deviation of a gaussian distribution), 3 times larger than the desired length increment.

These data show that the sizes of the central nanoblocks cannot be controlled with a sufficient accuracy, which should be of 0.1 nm or smaller to be sure that the length uncertainty remains smaller than the theoretical length increment between adjacent nanoblocks. If this condition is not fulfilled, the phase values realized on the sample can be largely different from the desired ones, since the phase variation with the geometry is very steep. Moreover, there is a high risk to realize a non-monotonous and chaotic phase variation instead of a monotonous variation. In other words, the phase cannot be correctly sampled because of the high quality factor. The problem is even worse for the lenses: since the central Fresnel zones are composed of a few tens of nanoblocks (see Supplementary Table 2), the theoretical length increment between adjacent nanoblocks is probably smaller than 0.2 nm. Note that a tiny variation of any other geometrical parameter (sidewall tilt, undercut...) will have a similar detrimental and uncontrolled impact on the phase sampling.

This issue of phase sampling is an intrinsic limitation of wavefront shaping with high-quality factor meta-atoms. Even if the authors demonstrate an excellent control over the fabrication, it is not sufficient to perform wavefront shaping with good performance. There is a trade-off between the quality factor of the meta-atoms and the performance of wavefront shaping. In my opinion, this crucial point should not be underestimated. It is misleading for potential readers to promote wavefront control with high quality factors without discussing the intrinsic limitations of the approach.

The incorrect phase sampling is probably the main reason for the small efficiencies reported by the authors (<50% for the beam deflectors and 20% for the lens). More importantly, an incorrect phase sampling results in the presence of a lot of stray light. For the beam deflectors, undesired diffraction orders carry a large fraction of the transmitted power at resonance. In TE polarization, the zeroth order diffraction efficiency is on the order of 50%, a value that is roughly equal to the diffraction efficiency of the useful first order, see Fig. 1a. In TM polarization, it is worse since the zeroth order carries more power than the first order, see Fig. 1b. These values show

that the phase produced by the metasurface is significantly different from the desired one. Figure 4 also shows a lot of stray light around the focal spot of the different lenses.

I do not recommend publication of this manuscript in Nature Communications.

Response: We apologize if in our previous response to the reviewer's comment we have not fully addressed the particular concern of the phase sampling resolution. We agree with the reviewer that given current fabrication constraints there is an inherent trade-off between the quality factor of the meta-atoms and the accuracy of phase sampling, and hence wavefront shaping. To discuss this important limitation, we have added the following paragraph to the revised manuscript on page 13:

In wavefront shaping with high quality factors there is an inherent tradeoff between quality factor and accurate phase sampling due to fabrication limitations. This is due to the rapid variation of the phase on resonance. For example, in the structure reported here, the phase of the transmitted light varies with 9.4 rad/nm within the range of $\pi/2$ to $3\pi/2$ (see Fig. 1b). As a result, a precision of the nanorod side length of 0.17 nm is required to sample the phase at a $\pi/2$ increment in this range. Furthermore, this required precision generally becomes stricter when increasing the quality factor. Our fabrication process results in an accuracy of the nanoblock side length of less than 0.6 nm, which produces considerable errors in phase sampling, in turn causing the appearance of stray light in the focal plane of the fabricated metalenses (Fig. 4j). This shows that fabrication imperfections are currently a limiting factor in the demonstrated quality factors and diffraction efficiencies. We expect an improved performance with higher fabrication uniformity and lower side wall surface roughness. While there is still room for improving the electron beam lithography-based process implemented here, using methods such as scanning probe lithography or atomic layer etching may allow fabrication with near atomic layer accuracy.

Another approach is to employ different optical modes and unit-cell geometries that enable wavefront shaping with similar quality factors but with more relaxed fabrication requirements. For example, a configuration in reflection with similar quality factors shows a slower variation of the phase of 2.64 rad/nm requiring only a 0.6 nm precision to sample the phase at a $\pi/2$ increment in the $\pi/2$ to $3\pi/2$ range (see Supplementary Note 6). Furthermore, accounting for fabrication imperfections in the device design is also expected to result in better device performance. This could be done by creating a new iteration of the phase look up table (Fig. 1c) by considering the fabrication imperfections identified in Supplementary Fig. 2.

We believe that with this revision to the manuscript, this important point noted by Reviewer 2 regarding the fabrication accuracy vs. phase sampling trade-off is now more clearly detailed in the manuscript. We hope that this will provide useful context for the reader. Notably, we emphasize that even with these limitations, our results markedly exceed the state of the art in all key performance metrics of wavefront shaping (quality factor, numerical aperture, oblique illumination performance, polarization independence). As a result, we continue to be convinced that our work deserves dissemination in a high impact journal to spark further developments in the field.

Additional comments:

- 2. Comment:** In the revised version of the Supplementary Information, the authors have added the measurements of the transmission spectrum of the lenses (Sup Fig 6). The authors should define what they mean by "transmission" and "diffraction efficiency"

(Fig. 4e and Sup Fig 7). In particular, the difference between the two quantities should be clearly indicated. Concerning the measurement of the diffraction efficiency, the authors should explain how they separate the different diffraction orders.

Response: We thank the reviewer for this important comment regarding the characterization of the metasurfaces. To address this comment, we have added the following paragraph to the methods section of the revised manuscript on page 17:

The transmission of the metasurfaces, $T = P_T/P_i$, is calculated by recording the power transmitted through the metasurface, P_T , normalized by the power incident on the metasurface, P_i . For the beam deflectors, the diffraction efficiency is defined as the fraction of transmitted power coupled into a specific diffraction order. To this end the intensity around each diffraction order is integrated within a square with a side length of 4 FWHM of the intensity of the diffraction order. For the metalenses the diffraction efficiency is defined as the fraction of transmitted power coupled into a circle around the focal spot with a radius of 2 times the airy disk radius.

3. **Comment:** The authors have added in the Supplementary Information important details about the realization of the deflectors and lenses (Sup Tables 1 and 2). Given its importance, the information about the size of the devices (150 μm for the deflectors and 100 μm for the lenses) should be included in the main text. The fact that the lens design is based on a parabolic phase profile should also be included in the main text.

Response: In response to this comment, we have added the following sentence to the revised manuscript on page 12 noting that a paraboloidal phase profile is used for the lens design:

The metalenses are designed by imposing a paraboloidal phase profile on the transmitted light by setting the nanoblock size lengths according to the look up table in Fig. 1c.

Furthermore, we have added a sentence to the caption of Fig. 2, 3 and 4 to provide information about the device sizes. Additionally, we would like to refer the reviewer to the Methods section of the main text on page 18, where the information on size of the devices and the phase profile for the lens is also listed.

4. **Comment:** The calculation of the Strehl ratio of the lenses is done by integrating the intensity in the focal plane around the focal spot within a radius of 4 times the diffraction-limited Airy disk radius. Since the intensity spreads in the focal plane beyond 4 times the Airy disk radius (according to Fig. 4 and Sup. Fig. 7), a non-negligible fraction of light may be missed. The calculations of the Strehl ratio should be done by integrating the intensity in the focal plane within a larger radius.

Response: We thank the reviewer for pointing out this inaccuracy in the characterization of our structures. In response to this comment, we have adapted the calculation of the Strehl ratio to integrate the intensity within an area of 8 times the Airy disk radius. As a result of that we observe a decrease in the Strehl ratios of the characterized lenses due to stray light that is now more accurately captured in the calculation. To include this adapted calculation, we have modified the reported values in the revised manuscript on page 12 and in Supplementary Fig. 8.

In closing, we thank the reviewer for their valuable input regarding the limitations of our work with respect to fabrication constraints, which have helped us to improve the quality of our presentation. We now hope to have addressed their concerns with the new discussion and analysis in the revised manuscript.

Reviewer #3:

In this paper the authors have used localized high-Q resonances in Mie-metasurfaces for linear optical components that can work over a narrow wavelength range. Although most of the community aims to achieve high performance optical functionality over a wide wavelength range, there are cases where high performance optical meta-optics over a narrow bandwidth could be beneficial. The fabrication of these metasurfaces is impressive, the results are good and the work shown in the main manuscript and its supplemental is very thorough. There are several issues though that range from fundamental to minor, and they are described below:

1. **Comment:** To set the record straight, achieving high-Q resonances using interference of two modes was pioneered by A. Kodigala, T. Lepetit, Q. Gu, B. Bahari, Y.F. Nature, and 2017, "Lasing action from photonic bound states in continuum," Nature 541, 196–199 (2017) in the context of photonic crystals. Later, Kivshar et al. used this concept with Mie-metasurfaces and termed it super-cavity modes in a number of papers (see for example: M.V. Rybin, K.L. Koshelev, Z.F. Sadrieva, K.B. Samusev, A.A. Bogdanov, M.F. Limonov, and Y.S. Kivshar, "High-Q Supercavity Modes in Subwavelength Dielectric Resonators," Phys Rev Lett 119(24), 243901 (2017), K. Koshelev, S. Kruk, E. Melik-Gaykazyan, J.-H. Choi, A. Bogdanov, H.-G. Park, and Y. Kivshar, "Subwavelength dielectric resonators for nonlinear nanophotonics," Science 367(6475), 288–292 (2020)). Please note that although Kivshar et al. termed these modes also BICs or quasi-BICs, the physics is exactly the same as what this papers uses, i.e., using interference of two modes in individual resonators rather than non-local modes. Even the concept of placing these resonators on top of a conducting surface was already mentioned in these papers. None of this prior work has been cited in the paper which leads to the wrong impression that the authors have invented a new, non-local way of achieving high-Q states/resonances that can be used for optical components. Granted, most of the prior literature on "super-cavity modes" has dealt with lasing or nonlinear optical phenomena, but the physics is the same and the concept can be applied to linear optical phenomena too. Maybe the authors of this manuscript missed this prior work because it was always portrayed using a mixed language that included "BIC modes". I think that this is a glaring omission and the claims of novelty should be scaled accordingly.

Response: We thank the reviewer for bringing up the important and highly relevant concept of extended supercavity modes (A. Kodigala *et al.*) and local supercavity modes as pioneered by Kivshar and co-workers. As suggested by the reviewer we have added the references on supercavity modes (newly added Ref. 11 and 40, Koshelev *et al.* was already referenced in the initially submitted manuscript) to the revised manuscript on page 4:

Notably, interference of multiple extended¹¹ or local^{13,40} modes has been a common strategy to achieve a high quality factor optical response.

Additionally, we also added the reference on extended supercavity modes (newly added Ref. 11) to the following sentences on page 2 in the revised manuscript:

With optical micro- and nanostructures including Fabry-Pérot cavities^{1,2}, whispering gallery mode resonators^{3–6}, photonic crystals^{7,8}, guided mode structures⁹, and bound states in the continuum (BIC)^{10–13}, quality factors of up to 10^8 have been demonstrated. The high level of field enhancement and confinement attained in these structures has led to many advances in sensing^{14,15} active optical devices^{16,17}, light sources^{11,18} and amplification of photon-matter coupling^{2,19}.

As a further response to this comment, we performed a new, additional analysis of ED/EO mode and the light scattering of our structure in the context of local supercavity modes, which is described in the **newly added Supplementary Note 2**. Our analysis suggests that the physics reported here is different from the work on local supercavity modes, as explained in the following points:

- Localized supercavity modes are observed in a single isolated particle where two interfering modes strongly couple and result in a characteristic anti-crossing behavior observed in the scattered light. At a specific point in the mode splitting, the lower wavelength mode exhibits a maximum quality factor due to a destructive interference of the two mixed modes outside of the resonator (Rybin, M.V. *et al.*, 2017. *Physical review letters*, 119(24), p.243901). To analyze whether this occurs in our structure, we performed a similar analysis as in Rybin, *et al.*, and simulate the scattered light from our metasurface with varying aspect ratios with finite difference time domain simulations using a total field scattered field source. **Newly added Supplementary Fig. 16a** shows the total scattered light for different nanoblock side lengths and a fixed nanoblock height. **Newly added Supplementary Fig. 16b and c** show the quality factor and Fano parameters as extracted with a Fano fit for different nanoblock side lengths. As illustrated in the scattered light spectrum, the resonance wavelength of the ED/EO mode linearly shifts with varying aspect ratio and no anti-crossing of modes is observed. Additionally, we do not observe mode splitting or destructive interference of the two ED/EO modes.
- At the supercavity condition, the light scattered from the resonator shows a narrow resonance with a quality factor that increases rapidly with the tuning of a specific geometric parameter. At this condition the quality factor is maximized and the Fano resonance of the scattered light collapses as indicated by the divergence of the Fano parameter (see Rybin, M.V. *et al.*, 2017. *Physical review letters*, 119(24), p.243901.). This corresponds to the uncoupling of the resonance from the continuum. As illustrated in newly added Supplementary Fig. 16b and c, we observe a slow variation of the quality factor with a change in aspect ratio and a finite Fano parameter in the range of -0.8 to -0.4. This suggests that the physics of the ED/EO mode is different from the localized supercavity mode.
- Previous works on local supercavity modes, as highlighted by the reviewer, for example on lasing (Mylnikov, V. *et al.*, 2020, *ACS Nano*, 14(6), pp.7338-7346) or on nonlinear optical phenomena (Ref. 13, Ryabov, D. *et al.*, 2022. *Nanophotonics*, 11(17), pp.3981-3991., Panmai, M. *et al.*, 2022. *Nature Communications*, 13(1), p.2749.) report on these modes in a single isolated resonator on a substrate. While the same physics may be encountered in a sub-diffractive array of resonators, where there is significant inter particle coupling, we are unaware of any report on this to date. Therefore, the specific case of the sub-diffractive array of our structure further sets our work apart from the previous work mentioned by the reviewer.
- Notably, local supercavity modes are just one out of several types of high-Q resonances due to interference of Mie modes. For example, metasurfaces relying on toroidal modes (Ref. 45), electromagnetically induced transparency (Ref. 46), interference of multiple Mie modes (Hähnel, D. *et al.*, 2023. *Light: Science & Applications*, 12(1), p.97, Prokhorov, A.V., *et al.*, 2022. *ACS Photonics*, 9(12), pp.3869-3875) or anapole modes (Tripathi, A. *et al.* 2021. *Nano Letters*, 21(15), pp.6563-6568.) show high quality factors.

The points raised above and the analysis in Supplementary Note 2, suggest that the ED/EO mode is rather an interference of two Mie resonant modes, which individually

show a high Q factor. Such an increase of quality factor with higher mode order is typically observed in Mie scattering (Bohren, C.F. and Huffman, D.R., 2008. John Wiley & Sons, New York). Hence, our approach is similar to a Huygens metasurface, where an ED and MD interfere, but in our case one of the modes is a higher order mode, namely the EO.

To address this comment and highlight our new analysis in the context of supercavity modes, we have added the following sentence on page 4 of the revised manuscript and refer to the several papers pointed out by the reviewer:

Analyzing the light scattered from the metasurface with varying nanoblock aspect ratios suggests that the ED/EO mode is different from extended supercavity modes¹¹ or supercavity modes reported in isolated nanoparticles^{13,40} (see Supplementary Note 2).

Below we provide the analysis of the ED/EO mode in the context of supercavity modes in the **newly added Supplementary Note 2**.

Supplementary Note 2: Distinction from supercavity modes

We further analyze the properties of the ED/EO mode and the light scattering of our structure in the context of local supercavity modes reported in isolated nanoparticles nanoparticles^{5,6}. We record the total light scattered by a metasurface with uniformly sized nanoblocks using finite difference time domain simulations. To this end, we employ a total field scattered field light source with periodic boundary conditions. Supplementary Fig. 16a illustrates the total scattered light for varying side length L of the nanoblocks and a fixed nanoblock height. Supplementary Fig. 16b and c illustrate the quality factor and the Fano parameter for varying L as determined from a Fano fit to the total scattered light. The data for the structure illustrated in Fig. 1b, L = 555 nm, is highlighted in bold. We observe a linear shift of the ED/EO resonance wavelength with a change nanoblock aspect ratio. As compared to local supercavity modes, the ED/EO mode does not undergo a mode splitting and anti-crossing. Furthermore, we observe a slow variation of the quality factor with variation of the nanoblock side length and a Fano parameter within the range of -0.8 to -0.4. Conversely, for a supercavity mode the quality factor of the scattered light varies rapidly as a function of a geometric parameter and the Fano parameter diverges to infinity at the supercavity condition. Additionally, reports on supercavity modes to date observe scattering from an isolated nanoparticle, whereas the nanoblocks in our structure are arranged in a sub-diffractive array, where there is significant coupling between neighboring elements. This suggests that our concept of a higher order Mie resonant metasurface is different from the previously reported localized supercavity modes.

Supplementary Figure 16 | Light scattering from ED/EO metasurfaces. **a**, Simulated total scattered light from the higher order Mie resonant metasurfaces with $H = 695$ nm and $P = 736$ nm and $L = 510\text{--}570$ nm. Curves are displaced vertically by 2.5 for better visibility. Quality factors (**b**) and Fano parameters (**c**) extracted from the total scattered light in (**a**) with a Fano fit.

2. **Comment:** As prior reviewers have noted, the authors don't provide a clear path on how to design these high-Q resonances, other than numerical optimization. The lowest Mie modes of the resonators can be approximately calculated using analytical formulas, but the higher order modes are very sensitive to the detailed geometry of the structures. Moreover, the phase gradient numbers quoted in table S1 and used in the design of the metaoptics have to depend on the Q of the spectral features. But there is no way to know a-priori what this Q will be since this value will depend on a myriad of factors. A big part of scientific publications has to do with reproducibility, so how exactly could another group replicate these results? Does one first have to go through a fabrication step to figure out the Q's and parameter sensitivity, and only then one can design a phase profile using these resonances?

Response: We thank the reviewer for pointing to this important issue regarding the design method and reproducibility of our work. We apply the traditional forward design approach of designing our metasurface as applied in many of the seminal metasurface works (see Refs. 23, 29). In this work, we adopted a fairly conventional strategy which was to perform finite difference time domain simulations to create a lookup table for phase and amplitude (see Fig. 1c), and choose the nano block sizes according to this. The reviewer is correct in pointing out that in the fabricated structures, the phase and the Q will depend on a myriad of factors. As with most nanophotonic structures, there is generally a difference between the simulated target structure and the fabricated structure; hence it is difficult to know the device performance (transmission, phase ect.) a-priori. That is also indeed the case in our study. In our structures, these differences include tilting of the sidewalls, structure undercut, surface roughness and others (see Supplementary Fig. 2). In our design we did not account for these differences and their effect on parameters such as quality factor, phase, and transmission, hence the differences between Fig. 1b and Fig. 2b, see also Supplementary Fig. 2. *Nonetheless, using this fairly conventional approach combined with good control of the fabrication, we were able to design metalenses with phase gradients suitable for focusing at different numerical apertures.* We believe that in the future, by performing an iterative

design approach with feedback from the fabricated devices, the performance of the devices could be further increased. To address this comment and clarify this point to the reader we have added the following sentence to the main text on page 9,

The nanoblock size lengths are set based on the phase look up table determined from numerical simulations in Fig. 1c, by applying a traditional forward design.

and on page 12,

The metalenses are designed by imposing a paraboloidal phase profile on the transmitted light by setting the nanoblock size lengths according to the look up table in Fig. 1c.

and on page 13,

Furthermore, accounting for fabrication imperfections in the device design is also expected to result in better device performance. This could be done by creating a new iteration of the phase look up table (Fig. 1c) by considering the fabrication imperfections identified in Supplementary Fig. 2.

3. **Comment:** Angular dispersion: the authors show very nice results of low angular dispersion which in turn make the design of metaoptics possible. However, there is a claim in several parts of the paper that this is a feature of Mie resonant metasurfaces. I would like to point out that this is not a universal statement. When the meta-atom has a design that is close to isotropic (for example, close to cube or sphere), then the modes excited by plane waves at different angles of incidence will show low angular dispersion. However, the modes of a metaatom that deviates from this 1:1 aspect ratio will show a very large angular dispersion (think for example of a flat cylinder or prism as an extreme case).

Response: We thank the reviewer for their important comment regarding angular dispersion of optical metasurfaces. We would like to clarify that with *angular dispersion* we are referring to a change in the resonant wavelength vs. incident angle. We agree with the reviewer that for certain morphologies the optical response of Mie resonators can vary when illuminated at different angles. For example, for a meta-atom that deviates from the 1:1 aspect ratio, as the reviewer mentions, the change in the optical response vs angle can arise due to varying excitation of the modes of the three different axes. This is a result of the varying projections of the incident field along the respective axes, and does not necessarily arise from the angle dependent resonance wavelength. Specifically, for localized resonances, such as local surface plasmon resonance or Mie scattering in a sub-diffractive array, the angle-dependent resonance wavelength shift (angular dispersion) is dominated by interparticle interactions (see Zhang, X., *et al.*, 2020. *Light: Science & Applications*, 9(1), p.76.) and less by the particle shape.

To address this comment, we added the following clarifying remark on page 5 of the main text:

The observed flat angular dispersion (resonance wavelength vs. incidence angle) of our surface indicates that the studied high-Q mode is localized.

Lastly, we would like to clarify that we do not claim that low angular dispersion is a feature of Mie resonant metasurfaces. Specifically on page 5 of the main text we note, 'The observed flat angular dispersion (resonance wavelength vs. incidence angle) of our surface indicates that the studied high-Q mode is localized. Furthermore, the dispersion is of similar order as reported in low-order Mie-resonant metasurfaces based on the spectral overlap of an electric and magnetic dipole^{43,44}.' Hence, we point

out that low angular dispersion is a feature of a localized resonance, such as in the extreme example of a dipole.

4. **Comment:** If the claim that the spectral feature used in the paper is due to interference of the radiation pattern of two modes of an individual resonator, then I don't understand why the authors don't just show that in a simulation, instead of dedicating an entire section of the supplemental to mode volume, studying the spectral changes when one perturbs one resonator in an array, etc. Why not just perform a simulation with one resonator with absorbing boundaries and no periodic boundary conditions?

Response: We thank the reviewer for this valuable suggestion to study the radiation pattern of an individual resonator of our high-Q metasurface. As a response to this comment, we added a plot of the radiation pattern of an isolated nanoblock in free space at the electric octupole resonance to **newly added Supplementary Fig. 12c**. As can be seen from the multipole expansion on the isolated nanoblock in Supplementary Fig. 12a and 12b, the scattering at that wavelength is dominated by the EO mode, including a significant scattering from the ED and MD modes. Conversely, in the array we observe that both EO and ED are of similar strength, see Supplementary Fig. 14. We attribute this difference in the response of an individual scatterer compared to the response of the scatterer in the array, to the nearest neighbor interaction. As a result, we believe the radiation pattern of the individual scatterer cannot be directly transferred to the mode interference occurring in the array.

Supplementary Figure 12 | Multipole expansion of an isolated nanoblock. **a**, Scattering cross section of a single amorphous silicon nanoblock in free space with $L = 555 \text{ nm}$ and $H = 695 \text{ nm}$ calculated using the multipole expansion method¹. For comparison, the sum of the multipoles and the scattering cross section as calculated by FDTD is shown. **b**, A zoomed in view of (a) over the wavelength range of the electric octupole mode. **c**, *Radiation pattern of an isolated nanoblock in free space at the EO resonant wavelength $\lambda = 1.16 \mu\text{m}$.*

5. **Comment:** some of the choice of language is a bit problematic. Why do the authors choose to use “higher-order Mie resonant metasurface”? All this paper is showing is how to use the interference of two Mie modes for optical components. Same for the use of “ED/EO mode” (or maybe use “supercavity mode” as it was used in the past).

Response: We thank the reviewer for highlighting to the important issue of nomenclature of our mode/metasurface. In this manuscript we report on how to realize high quality factor optical components by using the interference of two Mie modes, where at least one of the modes is a higher order mode. As explained in response to

comment 1, using the term *supercavity mode* would be a misrepresentation. Our concept can be viewed as an extension to higher orders of a Huygens metasurface where ED and MD interfere. However, as the Huygens condition is not met in our structure, we believe the term *higher order Mie resonant metasurface* is more general and fits our concept the best.

6. **Comment:** why does the Q-factor increase when there is nearest neighbor coupling? Usually coupling leads to lowering of the Q.

Response: We thank the reviewer for highlighting this possibly counter intuitive optical response. We attribute the increase in quality factor when going from a single particle to an array to the enhancement of the electric dipole through multipole coupling of the ED and EO over the first several neighboring particles in the array. Such coupling between ED and EO has been reported in previous work (newly added Ref. 42: Prokhorov, A.V., *et al.*, 2022. *ACS Photonics*, 9(12), pp.3869-3875)), resulting in higher quality factor in an array as compared to the isolated particle. As opposed to Ref. 42, where the coupling induces a suppression of the ED, we observe an enhancement of the ED through coupling in the array.

To make this point clearer in the manuscript, we have added the following note on this to the main text on page 4-5:

For the ED/EO mode, we attribute this increase in quality factor with increasing array size to an enhancement of the ED through multipole coupling of the ED and EO in the array through neighboring particles, as similarly observed previously in a different geometry⁴².

7. **Comment:** Minor issue: several of the graphs are very small and have traces with overlapping lines & colors that can't be discerned.

Response: We thank the reviewer for this valuable comment regarding the illustrations of the manuscript. As a response to this comment, we carefully looked though all the illustrations and ensured good visibility of different traces and colors. In particular, to improve the visibility we have added insets to Fig. 3a and b, Supplementary Fig. 3 and 4, and we have improved the visibility of the data points in Supplementary Fig. S8f.

In summary, I think that this paper shows very nice and impressive results but the novelty claim should be recalibrated due to the glaring omission of prior work that was used in the context of other optical phenomena. The issues that I raised are serious enough that warrant a new version of the manuscript.

In closing, we thank the reviewer for their valuable input regarding design optimization and a relevant study that we had overlooked. We believe that in addressing their comments we have significantly improved the quality of the manuscript and its suitability for publication in Nature Communications.

REVIEWERS' COMMENTS

Reviewer #1 (Remarks to the Author):

I have gone through the comments and find the response satisfactory

Reviewer #2 (Remarks to the Author):

In the revised manuscript, the authors address in a satisfactory manner with a whole paragraph my main concern about the phase-sampling issue with high-Q resonances. The authors also answer to my other comments, and they revised the manuscript and supplementary information accordingly.

In my opinion, the revised manuscript is now suitable for publication, after a minor change (see last paragraph below).

I still feel that wavefront control is not the best application for the high-Q dielectric arrays realized in this work with an impressive fabrication accuracy. Other applications would not suffer from the same limitations. I am not sure that this work will have a large impact in the field of wavefront control with metasurfaces.

I have one last comment on the authors' response to my additional comment #1 about the difference between transmission and diffraction efficiency.

For the metalenses, the authors have defined the diffraction efficiency as a fraction of the transmitted power coupled into a circle around the focal spot. However, such a measurement does not allow to separate the diffraction orders, because they are not separated spatially in the focal plane. For instance, the zeroth diffraction order is identical to the incident beam and propagates unfocused. A fraction of its power is thus included in the circle around the focal spot used for the measurement. This is also the case for all other diffraction orders, whose wavefronts correspond to the transmission by lenses with different focal lengths (b.t.w. their superposition in amplitude is forming the stray light).

As a result, the measurement realized by the authors does not allow to evaluate the diffraction efficiency (for metalenses). The authors should change the name "diffraction efficiency" to a different name. Perhaps "Focusing efficiency", as used in some articles in the field, or something similar.

Reviewer #3 (Remarks to the Author):

I thank the authors for the detailed and thorough response to my review comments. My concerns have been addressed and I support the publication of this manuscript.

Response to Reviewer Comments

In this document the reviewer comments are pasted in blue, our response in black.

Reviewer #1:

I have gone through the comments and find the response satisfactory.

Response: We thank the reviewer for their valuable input in the review process and for deeming our response to their comments satisfactory.

Reviewer #2:

In the revised manuscript, the authors address in a satisfactory manner with a whole paragraph my main concern about the phase-sampling issue with high-Q resonances. The authors also answer to my other comments, and they revised the manuscript and supplementary information accordingly.

In my opinion, the revised manuscript is now suitable for publication, after a minor change (see last paragraph below).

I still feel that wavefront control is not the best application for the high-Q dielectric arrays realized in this work with an impressive fabrication accuracy. Other applications would not suffer from the same limitations. I am not sure that this work will have a large impact in the field of wavefront control with metasurfaces.

I have one last comment on the authors' response to my additional comment #1 about the difference between transmission and diffraction efficiency.

For the metalenses, the authors have defined the diffraction efficiency as a fraction of the transmitted power coupled into a circle around the focal spot. However, such a measurement does not allow to separate the diffraction orders, because they are not separated spatially in the focal plane. For instance, the zeroth diffraction order is identical to the incident beam and propagates unfocused. A fraction of its power is thus included in the circle around the focal spot used for the measurement. This is also the case for all other diffraction orders, whose wavefronts correspond to the transmission by lenses with different focal lengths (b.t.w. their superposition in amplitude is forming the stray light).

As a result, the measurement realized by the authors does not allow to evaluate the diffraction efficiency (for metalenses). The authors should change the name "diffraction efficiency" to a different name. Perhaps "Focusing efficiency", as used in some articles in the field, or something similar.

Response: We thank the reviewer for pointing out this important point regarding the measurement of the diffraction/focusing efficiency of our Metalenses. As a response to this comment, we have modified the term "diffraction efficiency" to "focusing efficiency" when referring to the metalenses throughout the manuscript, supplementary information and in all the relevant figures, as suggested by the reviewer. Specifically, these changes are now reflected in the revised manuscript on pages 9 and 13, in Figure 4 and in Supplementary Fig. 7.

We further thank the reviewer for supporting the publication of our revised manuscript.

Reviewer #3:

I thank the authors for the detailed and thorough response to my review comments. My concerns have been addressed and I support the publication of this manuscript.

Response: We thank the reviewer for highlighting the thoroughness of our response to their comments, and for supporting the publication of our manuscript.